# Genie: the first open-source ISO/IEC encoder for genomic data
Fabian Müntefering [1] ✉, Yeremia Gunawan Adhisantoso [1], Shubham Chandak [2], Jörn Ostermann[1], Mikel Hernaez [3] ✉ & Jan Voges [1] ✉

For the last two decades, the amount of genomic data produced by scientific and medical applications has been growing at a rapid pace. To enable software solutions that analyze, process, and transmit these data in an efficient and interoperable way, ISO and IEC released the first version of the compression standard MPEG-G in 2019. However, non-proprietary implementations of the standard are not openly available so far, limiting fair scientific assessment of the standard and, therefore, hindering its broad adoption. In this paper, we present Genie, to the best of our knowledge the first open-source encoder that compresses genomic data according to the MPEG-G standard. We demonstrate that Genie reaches state-of-the-art compression ratios while offering interoperability with any other standard-compliant decoder independent from its manufacturer. Finally, the ISO/IEC ecosystem ensures the long-term sustainability and decodability of the compressed data through the ISO/IEC-supported reference decoder.

Since the first sequencing of the human genome, the speed and efficiency of DNA sequencing have undergone dramatic improvements. This is why the amount of DNA sequencing data is expected to continue growing exponentially for the next years[1]. Further, the data generated by the sequencing machines passes through different processing steps that render the data in various formats with distinct statistical properties. Thus, new challenges in storing and processing of large volumes of genomic information continue to emerge. As a response, several specialized data compression algorithms that exploit the statistical properties of the various data formats have been developed to mitigate storage requirements via efficient encoding.

Specifically, the raw information returned by the sequencing machines consists of unordered records of nucleotide sequences annotated with record identifiers and nucleotide-level quality scores. They are commonly represented in the ASCII-based FASTQ format[2]. FASTQ files are then usually compressed with the general-purpose compressor gzip[3]. Specialized compression approaches for FASTQ data were developed and improved over the years; early and low-latency compressors (DSRC 2[4]) rely on separation and optimized encoding of the different data streams directly contained in FASTQ files (nucleotide sequences, record identifiers, quality scores). Later approaches improve the compression ratio by exploiting similarities between nucleotide sequences through record reordering (SCALCE[5]) or record assembly (Quip[6], LEON[7]). Recent compressors utilize a combination of reordering and assembly (FaStore[8], SPRING[9], PgRC[10],

Mstcom[11]) taking advantage of each approach while further improving compression performance. Some mentioned tools are full compressors for FASTQ-Files, including quality scores and record identifiers (e.g., FaStore[8], SPRING[9], Genozip[12], while others only focus on the nucleotide sequences and provide no functionality to encode the other data types (e.g., PgRC[10], Mstcom[11]).

Nucleotide sequences are then aligned to one or multiple reference sequences to reconstruct the most likely position of origin in the underlying genome. The generated aligned data is mostly stored in the binary BAM[13] format, which uses gzip as its compression engine. More specialized compression approaches for aligned data include DeeZ[14], CRAM 3.1[15] and Genozip[12]. These approaches explicitly exploit the redundancies between the mapped records and the reference sequence. Instead of encoding each mapped nucleotide sequence completely, it is sufficient to encode the mapping position, sequence length, and mismatches with the reference sequence.

While specialized compressors achieve excellent gains in compression ratio with respect to gzip[9,16], all solutions are incompatible with each other, hindering their broad applicability. Interestingly, ensuring interoperability while enabling specialized and continuous improvements is a well-known problem in audio and video coding. Leveraging years of developing highly successful standards[17–19], the Moving Picture Experts Group (MPEG), a working group of ISO and IEC, released its first international standard (ISO/

---

[1]Institut für Informationsverarbeitung (TNT), Leibniz University Hannover, Appelstraße 9a, Hannover 30167, Germany. [2]Department of Electrical Engineering, Stanford University, 350 Jane Stanford Way, Stanford, CA 94305, USA. [3]Center for Applied Medical Research (CIMA), University of Navarra, Av. de Pío XII, 55, Pamplona 31008 Navarra, Spain. ✉e-mail: muenteferi@tnt.uni-hannover.de; mhernaez@unav.es; voges@tnt.uni-hannover.de

IEC 23092, known as MPEG-G) for genomic information representation in 2019[20]. Among others, MPEG-G defines decoding processes aimed at improving storage and access to genomic information. Importantly, the encoding process is not standardized to allow implementation-specific improvements and innovations while maintaining full interoperability across alternative solutions. The standard currently comprises six parts (1. Transport and Storage of Genomic Information, 2. Coding of Genomic Information, 3. Metadata and Application Programming Interfaces, 4. Reference Software, 5. Conformance Testing, 6. Coding of Genomic Annotations). The processes implemented in the Genie encoder are mostly based on the second part of the standard. For more information about the standard and its features beyond the scope of this paper, we refer to the MPEG-G introductory paper[20]. Unfortunately, no open-source software solution implementing MPEG-G has been available so far, a situation that

(a)

(b)

**Fig. 1 | Genie architecture and parameters. a** Genie encoding process: the input data format is the uncompressed, binary MPEG-G record format that can store unaligned as well as aligned genomic data. FASTQ/BAM data must be transcoded to MPEG-G records before starting the encoding process. The records are regrouped into access units based on their alignment properties. Nucleotide sequences, record identifiers, and quality scores are then encoded into descriptor subsequences. To improve the efficiency of entropy encoding, a sequence transformation, splitting of symbols into subsymbols, and a subsymbol-level transformation are applied. Then the data is binarized into a stream of bits and compressed with CABAC. Finally, the compressed bitstreams and decoding parameters (collected during encoding) are wrapped into MPEG-G data structures. Optionally, these can be encapsulated into a container file together with external data (e.g. metadata or datasets encoded by third-party MPEG-G compliant software). We refer the reader to the methods section for a detailed description of all transformations. **b** Parameter optimization for the first subsequence of some selected descriptors (listed in Supplementary Table 1). Shown is the normalized compressed size, ordered from the worst (i.e., optimization progress of 0%) to the best (i.e., optimization progress of 100%) set of parameters found.

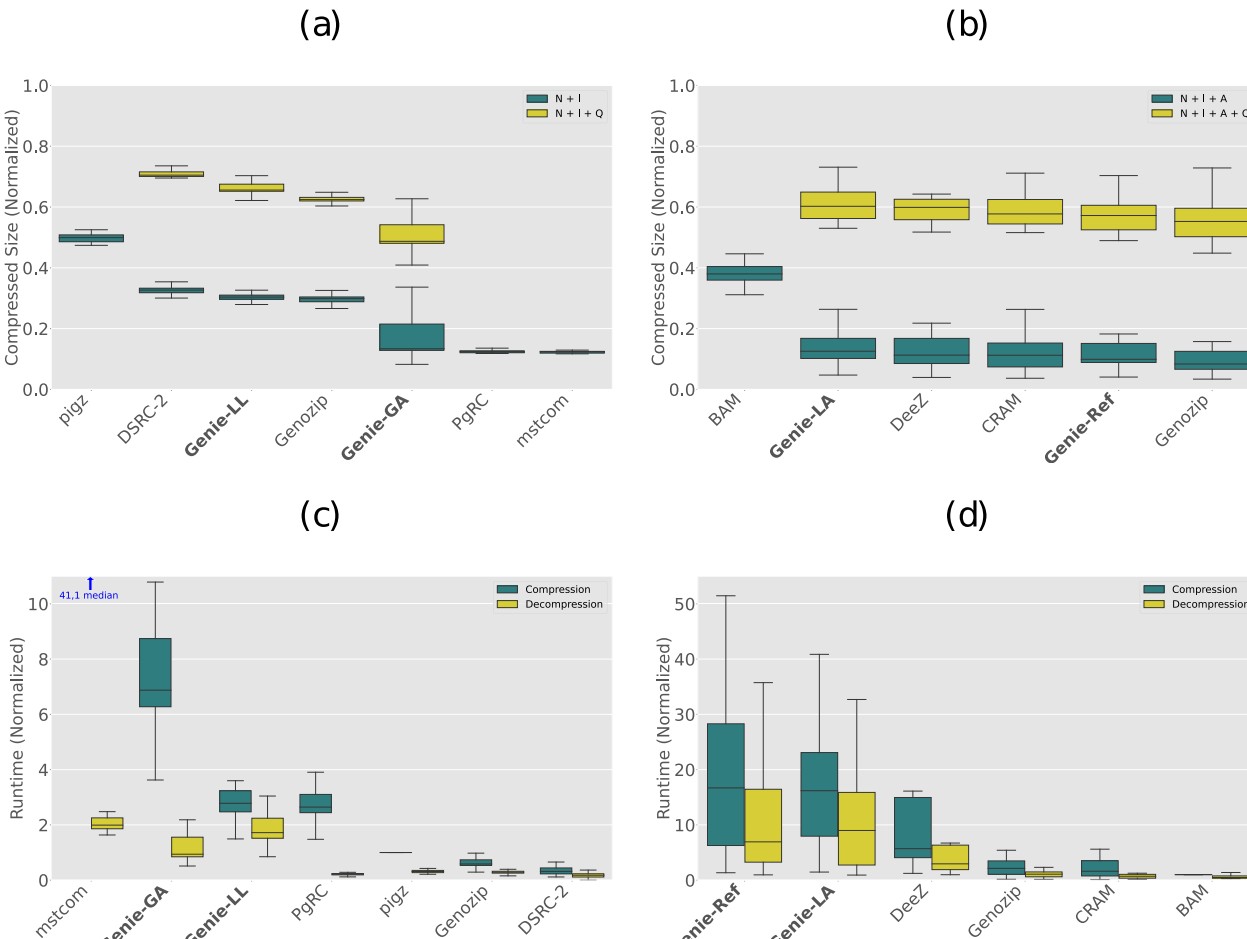

**Fig. 2 | Experimental compression ratios and runtime. a** Median compressed sizes of unaligned items (*n* = 57 items). The sizes are normalized in respect to gzip with quality scores. Note: the drop-in replacement pigz was used instead of gzip to make use of multi-threading capabilities. Q: Quality Scores. N + I: Nucleotide Sequences + Identifiers. **b** Median compressed sizes of aligned items (*n* = 23 items). The sizes are normalized in respect to BAM with quality scores. Q: Quality Scores, N + I + A: Nucleotide Sequences + Identifiers + Alignments, Genie-LL: Genie Low

Latency encoding, Genie-GA: Genie Global Assembly encoding, Genie-LA: Genie Local Assembly encoding, Genie-Ref: Genie Reference-Based encoding. **c** Median encoding and decoding runtime for unaligned items, normalized to the encoding time of gzip. The small arrow for Mstcom indicates a value outside the displayed range. **d** Median encoding and decoding runtime for aligned items, normalized to the encoding time of BAM. Whiskers in all panels show 1.5 times the interquartile range, boxes show quartiles 1–3, the center line shows the median.

dramatically hindered the adoption of the standard. In this paper, we present Genie, to the best of our knowledge, the first open-source encoder that produces MPEG-G compliant bitstreams. Genie combines existing encoding solutions such as SPRING[9] and GABAC[21] with newly implemented processes specified in the standard into a single framework, allowing to compress unaligned as well as aligned genomic data in an MPEG-G compliant manner.

## Results and discussion

### Encoding process

All parts of Genie are modular and exposed through interfaces. Therefore, the encoding process (Fig. 1a) can be modified and even extended with minimal changes to existing code. This allows Genie to be a future testbed for subsequent research on the topic of genomic data coding.

Genie groups genomic data into access units, the smallest independently decodable structures in MPEG-G. Records in an access unit must share specific alignment properties, such as the types of mismatches to the reference sequence. Multiple access units can be processed in parallel through multithreading, substantially boosting compression and decompression speed, or even be selectively streamed in network applications.

The information in the records is encoded as a set of so-called descriptors (listed in Supplementary Table 1). One descriptor represents one

aspect of the records and contains one or multiple subsequences of encoded data. Splitting up the information in this way is common practice in data compression algorithms and improves the statistics in the data streams[17], allowing a more efficient entropy encoding[21]. Genie supports multiple encoding strategies, depending on the availability of a reference sequence and alignment information.

1. *Reference-based encoding (Genie-Ref)* is selected automatically for aligned records if the reference sequence is provided. Nucleotide sequences are encoded by storing the alignment position, sequence length, and the edit operations (insertion, deletion, substitution) necessary for a reconstruction from the reference sequence.

2. *Local assembly encoding (Genie-LA)* is selected automatically for aligned records if the reference sequence is not provided. A buffer of recently encoded sequences is maintained, and a consensus reference sequence is computed by determining the most frequent nucleotide among the buffered sequences for each locus. That consensus reference is then used for reference-based encoding.

3. *Global assembly encoding (Genie-GA*, implemented by SPRING[9]) is selected automatically for unaligned records by default. A pseudo reference (i.e., not considering biological correctness) is created by searching for matching nucleotide sequences between all records in the dataset. The unaligned sequences are then aligned to the pseudo reference and used for reference-based encoding.

(a)

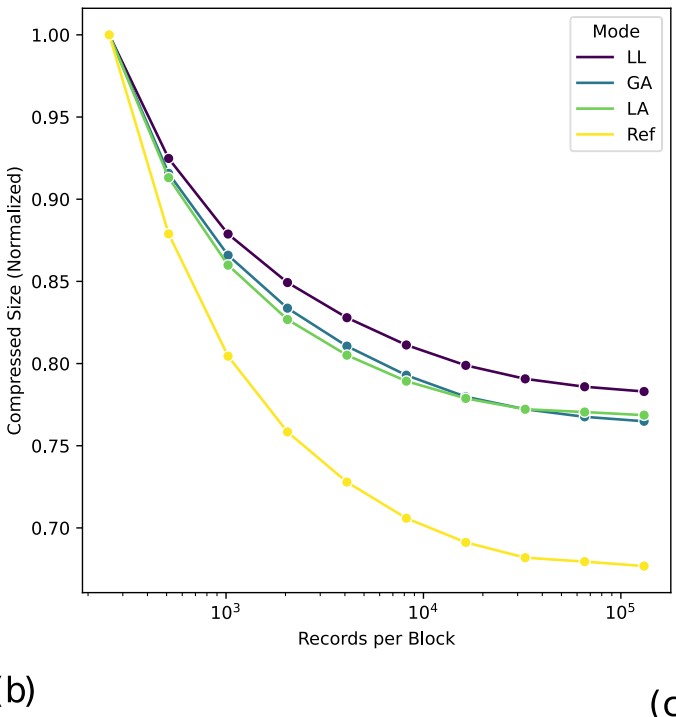

(b)                                                                                    (c)

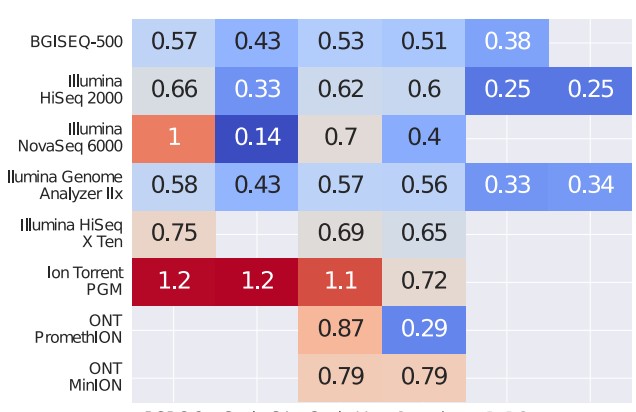

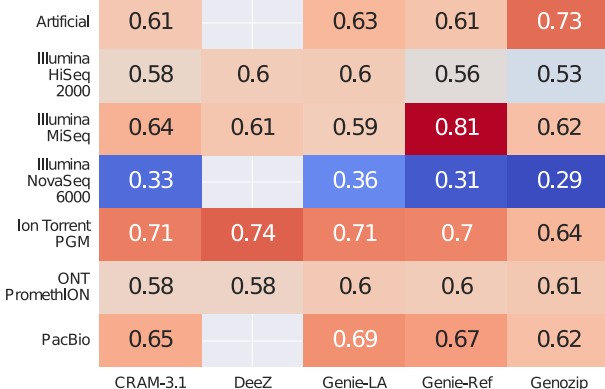

**Fig. 3 | Additional experimental data regarding block size and sequencing technology. a** Compressed size of the first $2^{18}$ records in item 38 (Ref, LA) or items 49 + 50 (GA, LL), respectively, depending on the encoding block size (smallest block size: $2^9$ records, largest block size $2^{18}$ records). **b**, **c** Compression ratios itemized by tool and sequencing technology. Normalized to gzip and BAM. Empty cells indicate items that are not supported by the tool or that processing failed.

4. *Low latency encoding (Genie-LL)* can be selected manually for unaligned records and encodes nucleotides verbatim instead of using any reference sequence.

Record identifiers are represented as delta-encoded, tokenized strings, exploiting the often prevalent naming patterns among records in a dataset. Supplementary Table 2 contains a list of available tokens. Quality scores in Genie are encoded verbatim or lossy through the compression scheme CALQ[22].

The descriptor subsequences generated by the previously mentioned encoding strategies are then entropy encoded with GABAC[21]. GABAC is based on various optional transformations that are followed by Context-Adaptive Arithmetic Coding (CABAC)[23]. The result of the encoding process in Genie is an MPEG-G compliant bitstream containing access units with genomic information, as well as the parameters needed for decoding. Genie optionally supports the encapsulation of bitstreams into an MPEG-G

container file that can provide additional features, such as the inclusion of metadata.

The Genie encoding process involves parameters that fall into two distinct categories. As described above, the first category encompasses parameters for the different encoding strategies responsible for generating descriptor sequences. The selection of these parameters is inherently determined by the characteristics of the data, such as whether the records are aligned or unaligned and the lengths of the nucleotide sequences within those data sets. Since these characteristics are directly data-dependent, optimization in this context is not necessary. The second category pertains to parameters associated with subsequent (optional) transformations of previously generated descriptor sequences and the final entropy encoding stage. Unlike the first category, these parameters are not rigidly dictated by the simple properties of the records. Instead, they provide flexibility and room for optimization to adapt to the statistical properties of the descriptor streams. Therefore, optimized parameters in the following context refers to

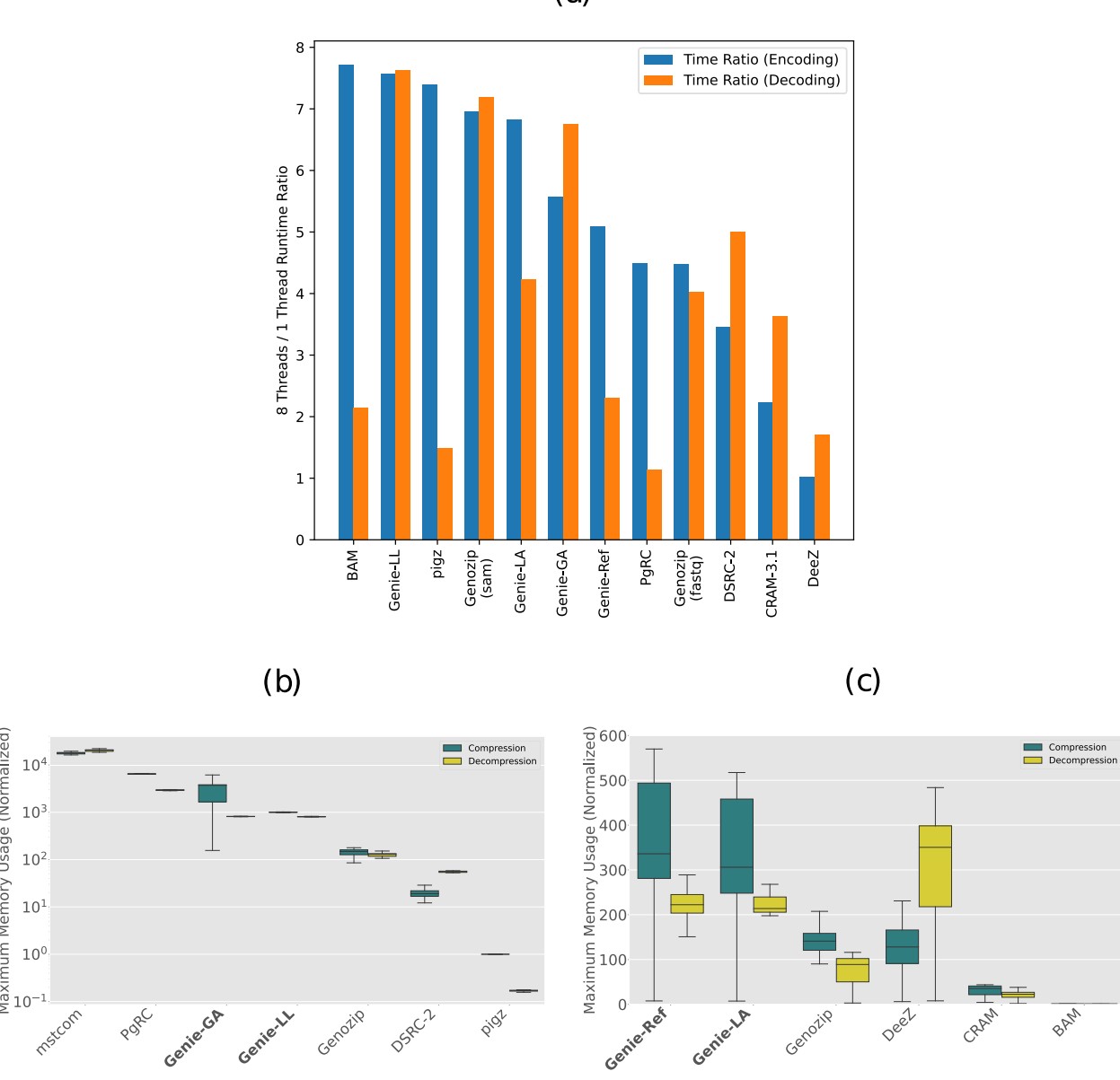

**Fig. 4 | Extended experimental data regarding multithreading and memory consumption. a** Ratio between single-threaded and multithreaded (8 threads) encoding/decoding times for item 01-1 (unaligned data) and item 9-2 (aligned data), respectively. **b** Median maximum memory usage for unaligned items (*n* = 57 items), normalized to the memory usage of gzip. Note that the scale is logarithmic because of the large difference in scale. The memory usage during compression (blue) is displayed as the left box and the memory usage during decompression (red) as the right box for each tool. **c** Median maximum memory usage for aligned items (*n* = 23 items), normalized to the memory usage of BAM. Whiskers in all panels show 1.5 times the interquartile range, boxes show quartiles 1–3, the center line shows the median.

the choice of transformations (sequence transformation, subsymbol transformation) and parameters of the entropy coding (binarization, context size, etc.) after descriptor subsequences have been created (see the Descriptor Subsequences arrow in Fig. 1a).

To compare the compression achieved by Genie to the state of the art, we used all available datasets from the MPEG-G genomic information database. Note that Genie in the current version does not support some aligned records with special properties (e.g., half-aligned records). For a fair comparison, these records were removed from all datasets or transformed into supported records prior to our experiments. For unaligned data, we compared gzip, DSRC 2[4], Genozip[12], PgRC[10], Mstcom[11] and Genie (low latency encoding, global assembly encoding). For aligned data, we compared BAM[13], CRAM 3.1[15], DeeZ[14], Genozip[12] and Genie (local assembly encoding, reference-based encoding).

**Parameter optimization**

To assess the reachable compression ratio of Genie, we created an optimized set of parameters by performing an exhaustive search. To achieve that, we generated a set of access units using four datasets (01-1, 2, 32, 37, more details available in Supplementary Table 3) from the MPEG-G genomic information database (https://mpeg.chiariglione.org/standards/MPEG-G/genomic-information-representation/MPEG-G-genomic-information-database-4) with all Genie encoding strategies and then chose those parameters that resulted in the best average compression. The inferred parameter set (shown in Supplementary Table 4) was then hard-coded for the entirety of the analysis. Figure 1b shows the normalized compressed size for a selected set of descriptors, ordered from the worst (i.e., optimization progress of 0%) to the best (i.e., optimization progress of 100%) set of parameters found. It can be concluded that parameter optimization can

impact the compression ratio substantially and, therefore, is crucial to reach acceptable results. We refer to Supplementary Data 1 for the final numeric results of all experiments. All numbers are distinct samples and were not measured repeatedly because of the computational requirements. During subsequent reruns of subsets of the simulations, we did not encounter any inconsistencies.

## Compression ratios

Figure 2a, b shows the achieved compression ratios of all benchmarked tools. The exact commands are documented in Supplementary Note 1. Note that the results of PgRC and Mstcom contain the compressed size of the record identifiers as computed in Genie, since these tools do not provide a way to encode this data themselves. For unaligned records, the reordering and assembly-based approaches generally outperform the other approaches, with Genie-GA, PgRC, and Mstcom all reaching a median compression ratio below 0.15 with respect to gzip, while the other approaches do not reach compression ratios below 0.25 with respect to gzip. Among the evaluated methods, Mstcom, PgRC, and Genie-GA were found to have the best median compression ratios, respectively. It should be noted that Genie-GA demonstrated a relatively higher degree of variance. Genozip outperforms both Genie-LL and DSRC-2, especially in quality score compression, while Genie-LL achieves better compression ratios than DSRC-2.

For aligned records, the difference in compression ratio between the benchmarked approaches (except BAM) is substantially smaller than that observed for unaligned records. It should be noted that Genie-LA does not use an external reference and therefore has to encode additional information. Thus, Genie-LA is not directly comparable to the other methods in the benchmark. Genie-Ref achieved a slightly better compression ratio than CRAM, while both are outperformed by Genozip.

Interestingly, while earlier benchmarks[16] showed a substantially better compression ratio for DeeZ in comparison to CRAM, we observed a similar compression ratio for both tools in our benchmarks. This is most likely attributed to the continuous development of the CRAM format over the last years, especially the introduction of new entropy codecs and specialized codecs for quality values and record identifiers in the latest version CRAM 3.1[15].

In Genie, the size of an access unit is a trade-off between granular access capabilities and compression ratio, as compression overhead as well as access granularity increase with smaller access units. Figure 3a shows the impact of access unit size on compression ratio in Genie. The compression ratio saturates as block sizes increase. The reference-based encoding mode (Genie-Ref) benefits most from larger block sizes, with a 32% decrease in compressed size between $2^9$ and $2^{18}$ records per access unit. The reason is likely that most of the record information is already located in the external reference, leading to smaller descriptor streams, thus more overhead for the same number of records compared to other encoding modes. For all other encoding modes, the impact of larger access units is similarly small, at approximately 23%.

## Compression runtimes

Figure 2c, d shows the median runtime required by each tool for the encoding and decoding processes. In general, the reordering and assembly-based approaches require more time during the encoding process compared to the other approaches for unaligned records. Mstcom is 40 times slower compared to the median runtime of gzip, while Genie-GA uses around 7 times and PgRC around 3 times as much runtime as gzip. Encoding of Genozip and DSRC-2 is even faster than gzip encoding, most likely due to reduced I/O requirements based on the substantially better compression rate those tools achieve compared to gzip. The PgRC, Genozip and DSRC-2 decoding runtimes are in the same order of magnitude as the gzip decoding process, while the decoding processes of the other tools took substantially longer, up to twice the encoding time of gzip. For aligned records, BAM, CRAM, and Genozip show comparable encoding speeds, while DeeZ encoding takes around 5 times as long. Genie-Ref and Genie-LA encoding runtimes are similar to each other, but approximately 3 to 4 times longer

than DeeZ. Further analysis shows that over 80% of the Genie runtime for aligned data is spent for the entropy encoding using CABAC. A plausible explanation for the high computational complexity of CABAC is the bit-based compression approach it employs, while most other entropy codecs process larger symbols. However, this issue could be resolved in future versions of Genie, as upcoming editions of the MPEG-G standard promise faster entropy encoding by including a number of low-complexity entropy codecs, such as BSC[24], LZMA[25], or Zstandard[26].

We also recorded the single-threaded performance of all tools for two selected data items to provide a qualitative comparison of the impact of multithreading. These results are reported in the extended data Fig. 4a. The benefit of multiple threads differs notably between tools. A plausible reason could be a difference in how coding speed is limited by CPU resources in comparison to I/O bandwidth.

## Memory consumption

The extended data Fig. 4b, c show the memory consumption of all encoding and decoding processes. For unaligned records, the memory consumption of the benchmarked tools spans over several orders of magnitude, with Mstcom requiring over $10^4$ times the memory resources of gzip. Genie-GA requires substantially more memory than Genie-LL due to the pseudo reference that Genie-GA constructs in memory. Both Genie-GA and Genie-LL require less memory than Mstcom and PgRC but substantially more than Genozip, DSRC-2, and gzip. The difference between the tools in memory consumption is considerably smaller for aligned records. Genie-Ref and Genie-LA use the most memory, around twice the amount of Genozip and DeeZ. Genie-Ref uses slightly more memory than Genie-LA because of the reference sequence that is kept in memory during the coding processes. CRAM requires only around a quarter of the memory of Genozip and DeeZ.

## Impact of sequencing technology

We also examined the achievable compression ratio of all tools, depending on sequencing technology. The results are shown in Fig. 3b, c. For unaligned records, compatibility between sequencing technologies and compression tools is restricted by properties such as record length. For example, Mstcom and PgRC require constant record length, and Genie-GA has a maximum compatible read length of 511 bases. Sequencing technologies that typically do not generate data complying to those restrictions are not compressible with those tools. Only Genie-LL and Genozip are able to compress datasets from all tested sequencing technologies in our benchmarks. For both Genie-LL and Genie-GA, the compression ratio for Illumina-based sequencing data is considerably better than for Ion Torrent data, for which the compression ratio does not outperform gzip. This is likely due to a combination of factors. Generally, there is less overlap between records in this sequencing technology, caused by longer record length and higher base error probabilities. This reduces the performance of assembly-based approaches such as Genie-GA. While the compression parameters in our benchmark are tuned towards Illumina data only, Ion Torrent sequencing data exhibits different statistical properties. Lastly, in the absence of a good statistical data approximation through parameter tuning and pseudo reference construction, the greater context size of gzip can lead to better generalization. Further parameter tuning could help to alleviate this issue.

For aligned records, the results of all methods are more comparable than for unaligned records. For Illumina NovaSeq sequencing data, a relatively better compression ratio can be observed across all methods. This performance is achieved through the quality score binning applied in the NovaSeq sequencing technology. In contrast, Ion Torrent exhibits a relatively worse compression ratio across all methods, which can most likely be attributed to a lower base quality and, therefore, a higher noise level in the data.

Note that on datasets that have not been used previously for the optimization of transformation and entropy encoding parameters as described above, Genie still achieves acceptable compression ratios if the underlying statistical properties and sequencing technology do not differ entirely. This indicates that an optimized parameter set can generalize to other datasets, and adapting it to new data is rather a matter of fine-tuning.

Additional gains in compression ratio can be expected with a more versatile way to determine parameter sets.

## Conclusions

In summary, Genie — the, to the best of our knowledge, first open-source encoder for MPEG-G compliant compression — reaches state-of-the-art compression ratios while at the same time facilitating the interoperability and sustainability provided by the ISO/IEC ecosystem.

## Methods

### Data

We used all available datasets from the MPEG-G genomic information database (https://mpeg.chiariglione.org/standards/MPEG-G/genomic-information-representation/MPEG-G-genomic-information-database-4) for our simulations. We refer the reader to Supplementary Table 3 for more information.

Genie, in its current version, does not implement all features of the MPEG-G standard needed to represent all alignments possible in BAM format. To allow a fair comparison to other compression approaches, we filtered or transformed aligned items containing the following features that Genie does not support:

1. The alignment information of half-aligned paired records (one nucleotide sequence aligned, the other unaligned) was removed, i.e. the records were converted to completely unaligned records.
2. Supplementary and secondary alignments were removed.
3. Optional tags were removed from the records. Optional tags in SAM files provide additional annotations on a per-record level, such as details from base calling, alignment processes, or information about mate records in paired sequencing. The usefulness of these tags largely depends on the specific downstream tasks and their requirements. Genie includes a script in its repository to perform that filtering. All relevant information left in the resulting files is compressible in a lossless way by all tools used in the experiments.

### Transcoding

The Genie encoding/decoding processes use MPEG-G records as uncompressed input and output formats. MPEG-G records (referred to by their file extension mgrec) are a flexible binary data structure that can represent unaligned as well as aligned genomic data. Other uncompressed formats like FASTQ or BAM must be transcoded into MPEG-G records before any encoding with Genie can be executed. In the case of aligned data, the Genie encoder expects the records to be ordered by the alignment position of their first nucleotide sequence. Transcoders that can convert BAM and FASTQ files into MPEG-G records are included in the Genie application. Records in BAM files must be sorted by record identifiers before the BAM transcoder is started (for example, by using samtools). The transcoder output is automatically sorted by alignment position and thereby directly suitable as input for Genie compression.

### Regrouping of records

As a first step of encoding, the records are regrouped into access units. An access unit is a set of jointly encoded records, and the smallest decodable unit defined in MPEG-G. Records in an access unit must share specific properties. Those properties are:

1. Record class: the class of an MPEG-G record describes the type of alignment. Available classes are P (fully aligned to a reference sequence without mismatches), N (fully aligned to reference and the only mismatches are substitutions with N, referring to an unknown nucleotide), M (fully aligned to reference with arbitrary substitutions), I (fully aligned to reference with insertions, deletions, or clipping), U (completely unaligned) and HM (record with two paired nucleotide sequences, one aligned and one unaligned).
2. The number of nucleotide sequences in the records, i.e., if the records consist of two paired nucleotide sequences or single nucleotide sequence.

3. The specific reference sequence that the nucleotide sequences in the records are aligned to, if any.

Records with the same set of those properties are put in a buffer together. With 5 classes of records (P, N, M, I, HM), 2 pairing configurations (paired/unpaired), and $n$ reference sequences, there are at most $5 \cdot 2 \cdot n$ buffers for aligned records. Additionally, with one class (U), 2 pairing configurations (paired/unpaired), and no references, there are at most 2 buffers for unaligned sequences. However, in most datasets, not all of these record types occur, thus some of these buffers typically remain unused. Once the buffer reaches a certain threshold size or no more input data is available, the buffer is flushed, and all records in it are encoded as one access unit. Multiple threads can encode multiple access units in parallel.

### Encoding of records into descriptors

Once an access unit is assembled, it is encoded into descriptor subsequences. An overview of descriptors used in MPEG-G is contained in Supplementary Table 1. Nucleotide sequences are encoded by comparing them to their reference sequence and storing only the information necessary to reconstruct the records from the reference sequence. In the case of no available external reference sequence, a reference can be computed from the genomic records themselves (local assembly encoding, global assembly encoding based on SPRING[9], or the nucleotide sequence can be encoded verbatim (low latency encoding) without exploiting the redundancy between overlapping records and their reference.

Record identifiers in MPEG-G are represented as delta-encoded, tokenized strings. In the first step, each record identifier is converted into a sequence of tokens independently. We refer the reader to the Supplementary Table 2 for a list of all tokens available. In the next step, the record ID of each record is compared token by token with the ID of the previous record. If possible, tokens are replaced with the delta token (indicating two different numbers at the same place in both IDs) or the match token (tokens are completely identical). Assuming that record identifiers in a dataset often follow a certain naming pattern (e.g., containing incrementing numbers or constant strings), this exploits redundancy between the record IDs. For each token position, a descriptor subsequence encoding the token types and further descriptor subsequences for parameters associated with each token type are created.

Quality scores in Genie can be completely discarded or encoded verbatim. It is also possible to employ a lossy compression scheme using CALQ[22], which uses genotype uncertainty to quantize the quality scores adaptively, such that subsequent analysis of the genomic data is minimally impacted.

### Sequence transformations and entropy encoding

The descriptor subsequences generated by the previously mentioned encoding schemes are finally entropy encoded. This is performed by GABAC[21], an entropy encoding solution consisting of various transformations that can be optionally carried out, followed by CABAC (Context Adaptive Arithmetic Coding)[23] encoding.

One of the following transformations is applied at symbol level for each descriptor subsequence, yielding one or multiple transformed descriptor subsequences:

1. No transformation: No transformation is applied to the descriptor subsequence.
2. Equality encoding: The descriptor subsequence is demultiplexed into two transformed descriptor subsequences. The first transformed subsequence contains a binary flag $F_i$ for each symbol $S_i$.
   $F_i = 1$ indicates that $S_i = S_{i-1}$.
   $F_i = 0$ indicates $S_i \neq S_{i-1}$ and that the next value in the descriptor subsequence shall be reconstructed from the second transformed subsequence. The second transformed subsequence contains a reconstruction value $R_j$ for the $j$th flag $F$ with value 0. $R_j$ is computed

such that:

$$R_j = \begin{cases} S_i, & \text{if } S_i < S_{i-1} \\ S_i - 1, & \text{if } S_i > S_{i-1} \end{cases}$$

3. Match encoding: The descriptor subsequence is demultiplexed into three transformed descriptor subsequences. A buffer of the last $B$ encoded symbols is maintained. It is determined if for an $l$ as large as possible, with $1 < l \le B$, the string of the next $l$ symbols matches with a substring of length $l$ in the buffer. If a match is found, the starting index of that substring in the buffer is appended to the first transformed subsequence and $l$ to the second transformed subsequence. If no match is found, 0 is added as length to the second subsequence and the current symbol $S_i$ to the third subsequence.

4. Run length encoding: The descriptor subsequence is demultiplexed into two transformed descriptor subsequences. For a guard value $G$, the descriptor subsequence is partitioned into the longest possible runs $(L_j, S_j)$ of consecutively equal symbols, with the length $L_j$ and the symbol $S_j$ of the $j$th run. Each length $L_j$ is then decomposed into the form $L_j = n_j \cdot G + r_j$ with $0 < r_j \le G$. Finally, for every run, $n_j$ symbols with value $G$ followed by one symbol with value $r_j - 1$ are appended to the first transformed subsequence, and one symbol with value $S_j$ is appended to the second transformed subsequence.

After the sequence transformation is applied, each symbol can be optionally split into smaller subsymbols. The size in bits of the subsymbols must be a factor of the symbol size in the transformed descriptor subsequence. For example, a sequence of 6-bit symbols can be split into a 3-bit subsymbol sequence with twice the amount of elements. After subsymbol splitting, for each of the subsymbols, one of the following transformations is applied:

1. No transformation: The sequence of subsymbols is copied to an identical sequence of transformed subsymbols.
2. Look-Up-Table transformation: A two (coding order 1) or three (coding order 2) dimensional table is generated in which the frequency of each subsymbol is sampled depending on the context of the previous one (coding order 1) or two (coding order 2) subsymbols in the sequence. For each context, the symbols are then ordered by their frequency, such that the most frequent symbol is at index 0 and the least frequent symbol at the highest index. Every subsymbol in the sequence is then substituted with its index in the table given the current context. The look-up tables needed to inverse this transformation are included at the beginning of the transformed subsymbol sequence.

In the next step, the transformed subsymbol sequences are converted into a stream of bits with one of the following binarizations:

1. Binary: The subsymbol is encoded as a binary number with $n$ bits
2. Truncated unary: The subsymbol $S$ is encoded as a run of $S$ one bits followed by a single zero bit. If $S$ equals the maximum in the possible range of values, the trailing zero is truncated.
3. Exponential Golomb: If $N$ is the number of bits necessary to encode $S + 1$ as binary number, the subsymbol $S$ is encoded as $N$ zero bits, followed by the binary representation of $S$. Finally, the binarized values are entropy encoded using the context-adaptive arithmetic coding algorithm (CABAC). The coding order of CABAC can be adjusted between 0 and 2.

## Determination of encoding parameters

A parameter set in MPEG-G contains one full configuration per descriptor subsequence for the previously described transformations. As the MPEG-G standard permits only 256 parameter sets per dataset, and datasets usually contain many more access units than 256, it is impossible to find an optimized parameter set individually for each access unit. There are many advanced strategies imaginable to use the available slots for parameter sets as

efficiently as possible, and those leave room for further research. For this paper, however, we decided to generate baseline results with a simple approach. That approach is to use just one constant, globally optimized parameter set for every access unit and every dataset. This implies that there is exactly one configuration per descriptor subsequence to optimize. As an additional experiment, we also compared this approach of using only one, globally optimized parameter set to using an individually optimized parameter set per access unit under the aspect of the sequence transformation. Our results indicate that the loss in compression ratio is insignificant (<1%) when choosing the simpler, global approach. The numeric results for this experiment are presented in Supplementary Table 5.

To find a well-working, global set of parameters, we encoded datasets 01-1, 32, 02, and 37 in the MPEG-G genomic database with all encoding processes in Genie (Reference-Based encoding, Local Assembly encoding, Global Assembly encoding, Low Latency encoding). We extracted the descriptor subsequences of the first 10 access units for each combination of dataset and encoding process (in the following referred to as file) before any transformation or entropy coding was applied (resulting in 80 access units in total). Ten access units were chosen as a sample for the full file, since we found that this sample size provides a balanced approach, offering a manageable computational complexity while also already encompassing most types of access units that appear in the file in significant amounts.

For every descriptor subsequence, we ran an exhaustive search and explored all possible combinations of supported parameters. For each combination of parameters, we recorded the final compressed size. The explored parameters consisted of the sequence transformation choice (none, equality encoding, match encoding, run length encoding) and the following parameters for each resulting transformed descriptor subsequence:

1. CABAC coding order (0, 1, 2)
2. Number of subsymbols per symbol for subsymbol splitting (1, 2, 4, 8)
3. Subsymbol transformation (No transformation, LUT transformation)
4. Binarization (binary, truncated unary, exponential Golomb)

## Choice of sequence transformations

We first evaluated the choice of the sequence transformation, as it alters the semantics and statistics of the generated transformed descriptor subsequences heavily. Because it is possible to compute the optimal sequence transformation for each descriptor subsequence directly, we chose an analytical approach over standard optimization algorithms (like genetic algorithms) that only approximate the optimal solution. We determined for each possible combination of access unit, file (combination of dataset and encoding strategy), descriptor subsequence, and sequence transformation the combination of parameters resulting in the smallest possible compressed size. With that experimental data, we performed the following calculation:

Let $F$ be the set of all input files, $T$ the set of all sequence transformations and $S$ the set of all descriptor subsequences. Let

$$\text{TrDeSu}(s, t, f, i) \mid s \in S, t \in T, f \in F, i \in \mathbf{N}, i < 10$$

refer to the tuple of demultiplexed transformed descriptor subsequences obtained by applying sequence transformation $t$ to descriptor subsequence $s$ in access unit $i$ in file $f$. Let $\text{minSize}(\text{TrDeSu}(s, t, f, i))$ refer to the smallest possible compressed size (as determined in the experiments) for that tuple with arbitrary compression parameters, apart from the already fixed choice of sequence transformation.

The smallest total compressed size for that tuple in all 10 access units available for a file $f$ is then:

$$\text{totalMinSizeTrDeSu}(s, t, f) = \sum_{j \in \mathbf{N}, j < 10} \text{minSize}\left(\text{TrDeSu}(s, t, f, j)\right).$$

Now for each combination of descriptor subsequence $s$ and file $f$, it is possible to compare the compressed sizes depending on the chosen sequence transformation $t$. The best sequence transformation $t_{\text{best}}$ for $s$

and $f$ is the one resulting in the smallest compressed size totalMinSizeTrDeSu$(s,t,f)$). Every other sequence transformation will result in a worse compressed size, i.e. a loss of compression ratio with respect to the optimum occurs. That loss factor can be calculated as:

$$\text{loss}(s,t,f) = \frac{\text{totalMinSizeTrDeSu}(s,t,f)}{min_{j \in T}(\text{totalMinSizeTrDeSu}(s,j,f))}.$$

We then computed the average loss factor for a specific descriptor subsequence and sequence transformation over all files:

$$\overline{\text{loss}}(s,t) = \frac{\sum_{j \in F}\text{loss}(s,t,j)}{\parallel F \parallel}.$$

The best sequence transformation for each descriptor subsequence globally can then be obtained by minimizing that average loss in compression ratio:

$$t_{best}(s) = \arg\min_{j \in T}(\overline{\text{loss}}(s,j)).$$

### Choice of remaining parameters
We observed that the set of remaining parameters remained strongly consistent among the best solutions with the smallest compressed size, given a fixed sequence transformation and descriptor subsequence. Therefore, we chose the remaining parameters by counting how often they appeared in the best solutions and chose the most frequently occurring ones. We refer the reader to Supplementary Table 4 for the results. The resulting parameters were hard-coded into Genie to be used for the corresponding descriptor subsequence.

### Reporting summary
Further information on research design is available in the Nature Portfolio Reporting Summary linked to this article.

## Data availability
All sequencing data used for our benchmark is available in the MPEG-G Genomic Information Database (https://mpeg.chiariglione.org/standards/MPEG-G/genomic-information-representation/MPEG-G-genomic-information-database-4) on request. A subset of the data has also been made available for direct download from various online repositories by their respective original authors. Supplementary Table 3 provides more details on the online availability of each dataset. The numeric experimental results and the source data behind the graphs in the paper can be found in Supplementary Data 1.

## Code availability
Genie is publicly available at https://github.com/MueFab/genie. The version used for this paper is also available via Zenodo[27].

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

## Acknowledgements
This work has been supported under the following grants: 1. 01EK2204F financed by the German Federal Ministry of Education and Research (BMBF) 2. PID2020-114394RA-C33 financed by MCIN/AEI/10.13039/501100011033 3. RYC2021-033127-I financed by MCIN/AEI/10.13039/501100011033 and the European Union "NextGenerationEU"/PRTR

## Author contributions

F.M. was the main contributor to the software, designed and conducted the experiments, and led the writing of the manuscript. Y.G.A. contributed to the software development, analyzed the experimental results, and assisted in writing the manuscript. S.C. conceived the SPRING compression method, provided the code, and guided its integration and the experimental setup. J.O. supervised the project and provided advice and guidance throughout the project. M.H. and J.V. supervised the project and conceived the idea of the Genie software and integral parts of the key algorithms for the Genie encoder. All authors discussed the results and contributed to the final manuscript.

## Funding

## Competing interests

The authors declare the following competing interests: J.V. and J.O. declare that they are listed as inventors on the following patents, parts of which are implemented in the Genie software: CN110915140B, EP3311318B1, EP3652862B1, US10938415B2. All authors declare that they contributed to the ISO/IEC 23092 standard series.
