## [Peer Review File · Communications Biology]

Reviewers' comments:

Reviewer #1 (Remarks to the Author):

The paper presents the open-source MPEG-G encoder namely genie. The architecture of genie was introduced and the encoder was compared with other related genomic data compressors. I have some concerns as follows:

1. The literature review on related work is insufficient.
2. The comparison studies were performed on limited datasets. It is unclear why only these datasets were used. I think a more systematic comparison should be conducted by using more various datasets.
3. The compared methods might not be state of the art. For example, DSRC2 and Quip were published ten years ago. Some new methods like PgRC, Mstcom, and genozip can be considered.

PgRC 1.2: <https://doi.org/10.1101/2020.05.01.071720>

Mstcom: <https://doi.org/10.1371/journal.pcbi.1009229>

Genozip: <https://doi.org/10.1093/bioinformatics/btaa290>, the latest version is available at <https://www.genozip.com/>, It supports the compression of both FASTQ and BAM.

4. In addition to compress ratio and the running time, the memory consumption of the software should be reported in the main text too.
5. The performance of genie actually is not very impressive to me. Specially, the compress ratios of genie are competitive with other methods, whereas it takes much more execution time than other methods. The authors would consider show the advantage of genie from different angles.
6. The authors state in Methods section, "Genie in its current version does not implement all features of the MPEG-G standard needed to represent all alignments possible in BAM format." Sounds like genie has not been well wrapped up and there is room for improvement.

Reviewer #2 (Remarks to the Author):

This paper describes an open source encoder/decoder implementation, genie, for compressing/uncompressing genomic data according to the MPEG-G standard. It presents the performance of genie wrt available compression methods that use both aligned and unaligned reads (with and without quality scores). While the tool is useful, the paper could be substantially improved by addressing the following.

1. Given that running time is an important measure featured in other papers, it is unclear why the authors present the running time results in the extended data.
2. It is interesting that CRAM now matches DeeZ's compression performance. This is likely due to CRAM's adaptation of the strategy of jointly compressing all reads mapping to a specific locus, which was introduced by DeeZ. This should be clarified in the paper.
3. While the paper describes MPEG-G well, if the reader is unfamiliar with the standard, s/he would have

a hard time following what is happening without a diagram of the standard. The reader could also be directed to the paper entitled “An introduction to MPEG-G” for further clarification.

4. The comparison of genie with FASTA/Q compressors should include FaStore, SCALCE, and Leon.

5. It is unclear how low latency, local assembly, and reference-based encoding is implemented in the paper. It is dependent or independent of the parameters in the genie architecture described in figure 1a?

6. In the first paragraph on page 4 where the mention of “an optimized set of parameters” appears, it should be mentioned what these parameters are referring to.

7. Why was Genie only tested on 8 datasets from the MPEG-G genomic information database and not all of them. There should be some justification for this mentioned in the paper because it might give the reader the wrong impression with respect to the experimental portion of the paper.

8. Again on page 5, the last paragraph, it should be noted what the parameter set is referring to.

9. On page 8, in filtering out of data, why are optional tags removed? Does that leave out important information? Also it is unclear whether other approaches remove these items.

10. On page 11, why is $i < 10$? In other words, how come there are 9 possible access units in file f? 6 records classes, 2 possibilities for number of nucleotide sequences, and two possibilities for presence or absence of a reference sequence. A supplementary note on what the 9 possible access units are would be nice as the reader might get confused about this number.

11. Should there be a disclosure statement similar to that in doi/10.1146/annurev-biodatasci-072018-021229 stating Mikel Hernaez and Jan Voges have contributed to the development of MPEG-G standard?

Reviewer #3 (Remarks to the Author):

The paper reports results of a open source implementation of a ISO/IEC 23092 compliant encoder. The paper is clearly written and reports the context and objective of the work and the optimizations that have been implemented in the current version of the encoder. A set of experimental results in terms of compression performance and processing speed are also reported versus other state of the art approaches. I thus recommend the paper for publication. Although the paper is well structured and complete in the description of the encoder implementation and achieved results some additions could improve the technical description and complete the reported results. For instance, it could be interesting to also report the single-threaded performance in addition to the multi-threaded performance. Another interesting discussion would be to better elaborate on the overall loss or gain in coding efficiency for using an optimal parameter set for each access unit versus using one optimized parameter set for all access units. it could be also interesting to add results on the performance of long-read (nanopore) data. Another interesting result would be to evaluate the trade-off between access unit size and compression ratio. It would also be worth mentioning in the text that the more recent edition of the standard also include a set of low complexity entropy coders that so far are not supported by the Genie implementation, but that could be used in future to optimize speed performance for some type of data such as quality values for instance.

Response to Reviewer Comments

Submission ID: COMMSBIO-23-1107-T
Title: Genie: The First Open-Source ISO/IEC Encoder for Genomic Data
Authors: Fabian Müntefering, Yeremia Gunawan Adhisantoso, Shubham Chandak, Jörn Ostermann, Mikel Hernaez, Jan Voges

Dear Editor and Reviewers,

We would like to thank you for your time and the constructive comments and suggestions that have helped to improve the quality of this manuscript. We have carefully addressed all comments and provide here a point-by-point response, including detailed descriptions. Additionally, we highlighted all the amends on the revised manuscript in blue color.

Please kindly note that in this major revision of our work, we invested significant time and multiple months of computing resources in additional simulations requested by the reviewers. We believe that these additional results allow the reader to evaluate Genie from multiple important angles. We hope that our revisions, along with our point-by-point response, satisfy all comments.

Yours sincerely,
Fabian Müntefering, on behalf of all authors

Reviewer 1

Comment 1.1 — The literature review on related work is insufficient.

Reply 1.1 — Thank you for your feedback regarding the literature review section. In response to your comment, we have expanded this section to include a broader range of data compression methods for unaligned genomic records. These additions include PgRC (1), Mstcom (2), Genozip (3), FaStore (4), SCALCE (5), and LEON (6), each referenced appropriately.

Moreover, we have provided a small overview of the different approaches they employ, categorizing them into distinct groups such as record reordering, record assembly, and combinations thereof. This categorization aims to offer a clearer understanding of the landscape in genomic data compression:

Specialized compression approaches for FASTQ data were developed and improved over the years; early and low-latency compressors (DSRC 2 (7)) rely on separation and optimized encoding of the different data streams directly contained in FASTQ files (nucleotide sequences, record identifiers, quality scores). Later approaches improve the compression ratio by exploiting similarities between nucleotide sequences through record reordering (SCALCE (5)) or record assembly (Quip (8), LEON (6)). Recent compressors utilize a combination of reordering and assembly (FaStore (4), SPRING (9), PgRC (1), Mstcom (2)) taking advantage of each approach while further improving compression performance. Some mentioned tools are full compressors for FASTQ-Files, including quality scores and record identifiers (e.g., FaStore (4), SPRING (9), Genozip (3)), while others only focus on the nucleotide sequences and provide no functionality to encode the other data types (e.g., PgRC (1), Mstcom (2))

For the portion of the literature review dealing with aligned genomic records, we have acknowledged Genozip (which supports the compression of both aligned and unaligned data) as a state of the art method, and we have also incorporated it into our experimental comparisons:

More specialized compression approaches for aligned data include DeeZ (10), CRAM 3.1 (11) and Genozip (3). These approaches explicitly exploit the redundancies between the mapped records and the reference sequence. Instead of encoding each mapped nucleotide sequence completely, it is sufficient to encode the mapping position, sequence length, and mismatches with the reference sequence.

Comment 1.2 — The comparison studies were performed on limited datasets. It is unclear why only these datasets were used. I think a more systematic comparison should be conducted by using more various datasets.

Reply 1.2 — Thank you for your comment. In response, we have significantly expanded our experimental scope to include a more diverse and extensive range of datasets. Specifically, we incorporated data from the complete MPEG-G genomic information database (GIDB) (12), which offers a broad spectrum of genomic datasets. These datasets span multiple species, including *H. sapiens*, *E. coli*, *S. cerevisiae*, *D. melanogaster*, *T. cacao*, *P. aeruginosa*, and ϕ X174, thereby covering a range of biological kingdoms, but still with a notable emphasis on human sequencing data.

Furthermore, the GIDB provides data from diverse sequencing technologies, including those developed by Illumina, Oxford Nanopore Technologies, Pacific Biosciences, Thermo Fisher Scientific (Ion

Torrent), and BGI. This variety ensures a comprehensive analysis across different sequencing technologies. Our revised experiments encompass both aligned and unaligned genomic records. By utilizing the full extent of the GIDB, we aim to offer a more systematic and thorough comparison, highlighting the strengths and limitations of various compression methods across different species and technologies.

In the supplementary section of our paper, we have included an extended table in appendix C that details all the datasets used, outlining their key properties and providing download links where available. Our dataset now consists of 61 unaligned and 28 aligned items, with a total of 89 distinct datasets. Our total set of experimental data now comprises 958 data points. Figures 2 and 3 in our paper have been updated to illustrate the distribution of results from this expanded dataset. We also introduced the new figures 3b and 3c, which segregate the median compression ratios of the compression tools by sequencing technology. It is important to note that not all tools were compatible with every dataset, such data points were excluded from further analysis. For instance, PgRC and Mstcom are limited to datasets with constant record lengths and are thus incompatible with nanopore sequencing datasets. The following description of our new results was added to the manuscript:

Figures 2a and 2b show the achieved compression ratios of all benchmarked tools. The exact commands are documented in supplementary material E. Note that the results of PgRC and Mstcom contain the compressed size of the record identifiers as computed in Genie, since these tools do not provide a way to encode this data themselves. For unaligned records, the reordering and assembly-based approaches generally outperform the other approaches, with Genie-GA, PgRC, and Mstcom all reaching a median compression ratio below 0.15 with respect to gzip, while the other approaches do not reach compression ratios below 0.25 with respect to gzip. Among the evaluated methods, Mstcom, PgRC, and Genie-GA were found to have the best median compression ratios, respectively. It should be noted that Genie-GA demonstrated a relatively higher degree of variance. Genozip outperforms both Genie-LL and DSRC-2, especially in quality score compression, while Genie-LL achieves better compression ratios than DSRC-2.

For aligned records, the difference in compression ratio between the benchmarked approaches (except BAM) is significantly smaller than that observed for unaligned records. It should be noted that Genie-LA does not use an external reference and therefore has to encode additional information. Thus, Genie-LA is not directly comparable to the other methods in the benchmark. Genie-Ref achieved a slightly better compression ratio than CRAM, while both are outperformed by Genozip.

Figures 2c and 2d show the median runtime required by each tool for the encoding and decoding processes. In general, the reordering and assembly-based approaches require more time during the encoding process compared to the other approaches for unaligned records. Mstcom is 40 times slower compared to the median runtime of gzip, while Genie-GA uses around 7 times and PgRC around 3 times as much runtime as gzip. Encoding of Genozip and DSRC-2 is even faster than gzip encoding, most likely due to reduced I/O requirements based on the significantly better compression rate those tools achieve compared to gzip. The PgRC, Genozip and DSRC-2 decoding runtimes are in the same order of magnitude as the gzip decoding process, while the decoding processes of the other tools took significantly longer, up to twice the encoding time of gzip. For aligned records, BAM, CRAM, and Genozip show comparable encoding speeds, while DeeZ encoding takes around 5 times as long. Genie-Ref and Genie-LA encoding runtimes are similar to each other, but approxi-

mately 3 to 4 times longer than DeeZ. Further analysis shows that over 80% of the Genie runtime for aligned data is spent for the entropy encoding using CABAC. A plausible explanation for the high computational complexity of CABAC is the bit-based compression approach it employs, while most other entropy codecs process larger symbols. However, this issue could be resolved in future versions of Genie, as upcoming editions of the MPEG-G standard promise faster entropy encoding by including a number of low-complexity entropy codecs, such as BSC (13), LZMA (14), or Zstandard (15).

We also examined the achievable compression ratio of all tools, depending on sequencing technology. The results are shown in figures 3b and 3c. For unaligned records, compatibility between sequencing technologies and compression tools is restricted by properties such as record length. For example, Mstcom and PgRC require constant record length, and Genie-GA has a maximum compatible read length of 511 bases. Sequencing technologies that typically do not generate data complying to those restrictions are not compressible with those tools. Only Genie-LL and Genozip are able to compress datasets from all tested sequencing technologies in our benchmarks. For both Genie-LL and Genie-GA, the compression ratio for Illumina-based sequencing data is considerably better than for Ion Torrent data, for which the compression ratio does not outperform gzip. This is likely due to a combination of factors. Generally, there is less overlap between records in this sequencing technology, caused by longer record length and higher base error probabilities. This reduces the performance of assembly-based approaches such as Genie-GA. While the compression parameters in our benchmark are tuned towards Illumina data only, Ion Torrent sequencing data exhibits different statistical properties. Lastly, in the absence of a good statistical data approximation through parameter tuning and pseudo reference construction, the greater context size of gzip can lead to better generalization. Further parameter tuning could help to alleviate this issue.

For aligned records, the results of all methods are more comparable than for unaligned records. For Illumina NovaSeq sequencing data, a relatively better compression ratio can be observed across all methods. This performance is achieved through the quality score binning applied in the NovaSeq sequencing technology. In contrast, Ion Torrent exhibits a relatively worse compression ratio across all methods, which can most likely be attributed to a lower base quality and, therefore, a higher noise level in the data.

Figure 2: **Genie compression ratios.** **a**, Compressed sizes of unaligned items. The sizes are normalized in respect to gzip with quality scores. Note: the drop-in replacement pigz was used instead of gzip to make use of multi-threading capabilities. QS: Quality Scores. $N + I$: Nucleotide Sequences + Identifiers. **b**, Compressed sizes of aligned items. The sizes are normalized in respect to BAM with quality scores. QS: Quality Scores. $N + I + A$: Nucleotide Sequences + Identifiers + Alignments. *Genie-LL*: Genie Low Latency encoding. *Genie-GA*: Genie Global Assembly encoding. *Genie-LA*: Genie Local Assembly encoding. *Genie-Ref*: Genie Reference-Based encoding. **c**, Encoding and decoding runtime for unaligned items, normalized to the encoding time of gzip. The small arrow for Mstcom indicates a value outside the displayed range. **d**, Encoding and decoding runtime for aligned items, normalized to the encoding time of BAM.

Figure 3: **Additional experimental data.** **b + c**, compression ratios itemized by tool and sequencing technology. Normalized to gzip and BAM. Empty cells indicate items that are not supported by the tool or that processing failed.

Comment 1.3 — The compared methods might not be state of the art. For example, DSRC2 and Quip were published ten years ago. Some new methods like PgRC, Mstcom, and genozip can be considered. PgRC 1.2: <https://doi.org/10.1101/2020.05.01.071720>
Mstcom: <https://doi.org/10.1371/journal.pcbi.1009229>
Genozip: <https://doi.org/10.1093/bioinformatics/btaa290>,
the latest version is available at <https://www.genozip.com/>,
It supports the compression of both FASTQ and BAM.

Reply 1.3 — We appreciate your suggestions and have incorporated PgRC, Mstcom, and Genozip into our extended set of experiments. The results are now reflected in the updated Figure 2 (see reply 1.2).

Mstcom and PgRC do not support encoding of record identifiers natively. To ensure a fair comparison, we adapted our approach by extracting the size of the compressed record identifiers from the Genie bitstream and added this to the final compressed size generated by Mstcom and PgRC. This adjustment allows us to maintain consistency in the amount of information compressed across all tools. We also added a remark to appendix E to declare that approach transparently:

Note: this tool does not support the compression of record identifiers. We added the size of the compressed identifiers obtained in the Genie (global assembly) experiments to the results of this tool to enable a fair comparison with the other tools.

A similar note was included in the main section of the paper:

Note that the results of PgRC and Mstcom contain the compressed size of the record identifiers as computed in Genie, since these tools do not provide a way to encode this data themselves.

Regarding Genozip, we noticed that it offers compression modes for FASTQ data that utilize an external reference. However, to maintain comparability with other tools in our study, which do not rely on a reference genome for FASTQ compression, we chose to exclude these modes from our analysis. We are confident that these updates will more accurately reflect the state of the art in genomic data compression.

Comment 1.4 — In addition to compress ratio and the running time, the memory consumption of the software should be reported in the main text too.

Reply 1.4 — In response to your suggestion, we have expanded our experimental setup to include comprehensive memory profiling. We employed Memory Profiler (16) to record the memory usage of each compression tool at one-second wall clock intervals. This information allowed us to calculate the maximum, average, and median memory consumption for each tool and dataset.

We have incorporated this new information into the manuscript. Specifically, the extended figures have been updated to display the distribution of maximum memory usage for each tool during both compression and decompression phases. Our analysis showed that the trends observed in the median and average memory usage correlated with those seen in the maximum memory usage. Therefore, we focused on the maximum memory usage only in our graphical representation. While memory consumption is a crucial metric, our findings suggest that runtime and compression ratio tend to attract more attention, leading us to feature memory consumption primarily in the extended data figures. We added the following text to the manuscript:

The extended data figures 4b and 4c show the memory consumption of all encoding and decoding processes. For unaligned records, the memory consumption of the benchmarked tools spans over several orders of magnitude, with Mstcom requiring over 10^4 times the memory resources of gzip. Genie-GA requires significantly more memory than Genie-LL due to the pseudo reference that Genie-GA constructs in memory. Both Genie-GA and Genie-LL require less memory than Mstcom and PgRC but significantly more than Genozip, DSRC-2, and gzip. The difference between the tools in memory consumption is considerably smaller for aligned records. Genie-Ref and Genie-LA use the most memory, around twice the amount of Genozip and DeeZ. Genie-Ref uses slightly more memory than Genie-LA because of the reference sequence that is kept in memory during the coding processes. CRAM requires only around a quarter of the memory of Genozip and DeeZ.

Figure 4: **Extended experimental data.** **b**, Maximum memory usage for unaligned items, normalized to the memory usage of gzip. Note that the scale is logarithmic because of the large difference in scale. The memory usage during compression (blue) is displayed as the left box and the memory usage during decompression (red) as the right box for each tool. **c**, Maximum memory usage for aligned items, normalized to the memory usage of BAM.

Comment 1.5 — The performance of genie actually is not very impressive to me. Specially, the compress ratios of genie are competitive with other methods, whereas it takes much more execution time than other methods. The authors would consider show the advantage of genie from different angles.

Reply 1.5 — Thank you for your feedback on Genie's performance, particularly concerning its execution time. We understand your concerns and would like to offer some insights into the factors contributing to this aspect of Genie's performance.

Our benchmarks (using the callgrind (17) profiler) indicate that approximately 85% of Genie's execution time is dedicated to entropy encoding using CABAC (Context-Adaptive Binary Arithmetic Coding). CABAC tends to be slow in comparison, as it works on the level of single bits instead of larger symbols. CABAC was the only entropy coder specified by the MPEG-G standard at the time of writing.

We also acknowledge the latest developments in the MPEG-G standard, which now permits the use of low-complexity entropy codecs, for example BSC (13), LZMA (14) or Zstandard (15). These codecs are symbol-based, instead of the bit-based approach of CABAC, and have the potential to markedly improve execution speeds due to their simpler computational requirements. However, these low-complexity entropy codecs are not yet implemented in Genie since they were only recently (2024) added to the latest edition of the standard. Their inclusion could potentially enhance Genie's processing efficiency, aligning it more closely with the execution times of other methods, while also reducing the complexity of the parameter sets.

Comment 1.6 — The authors state in Methods section, “Genie in its current version does not implement all features of the MPEG-G standard needed to represent all alignments possible in BAM format.” Sounds like genie has not been well wrapped up and there is room for improvement.

Reply 1.6 — Thank you for your comment. It's important to note that Genie is currently a research-focused platform and is not (yet) intended for professional environments. The MPEG-G standard is extensive, and while Genie doesn't implement all its features at present, this is in line with its role as a tool for exploring the standard's possibilities, rather than being a fully featured application for industry use. Future developments will adapt to the evolving needs and insights of the research community.

Reviewer 2

Comment 2.1 — Given that running time is an important measure featured in other papers, it is unclear why the authors present the running time results in the extended data.

Reply 2.1 — Thank you for your comment. In response to your feedback, we have now included the improved figures 2c and 2d that prominently display the running time results, and these figures have been moved to the main text of the paper. This change ensures that the running time data is immediately accessible, alongside other critical measures like compression ratios. Presenting these metrics together provides a more comprehensive and clearer overview.

Recognizing the significant increase in the number of datasets analyzed, we have also adapted our presentation of the execution time measurements to utilize box plots. Box plots offer a representation of the distribution of timings, including median values and variability. We refer to reply 1.2 for more details about the updated figures.

Comment 2.2 — It is interesting that CRAM now matches DeeZ's compression performance. This is likely due to CRAM 3 adaptation of the strategy of jointly compressing all reads mapping to a specific locus, which was introduced by DeeZ. This should be clarified in the paper.

Reply 2.2 — Thank you for your comment regarding the performance of CRAM 3 and its comparison to DeeZ. While CRAM adopting the read compression strategy of DeeZ sounds like a plausible explanation, we have reviewed the latest CRAM specifications (18) and change logs and found no mention of such an adaptation.

Therefore, we believe that the improvements in CRAM compression performance are more likely created by the new custom compression codecs and data transformations introduced in CRAM 3.1 (11), especially the fqzcomp and name tokenizer methods introduced in cram 3.1 (18). To clarify this, we added the following statement to the text of the main paper:

Interestingly, while earlier benchmarks (19) showed a significantly better compression ratio for DeeZ in comparison to CRAM, we observed a similar compression ratio for both tools in our benchmarks. This is most likely attributed to the continuous development of the CRAM format over the last years, especially the introduction of new entropy codecs and specialized codecs for quality values and record identifiers in the latest version CRAM 3.1 (11).

Comment 2.3 — While the paper describes MPEG-G well, if the reader is unfamiliar with the standard, s/he would have a hard time following what is happening without a diagram of the standard. The reader could also be directed to the paper entitled “An introduction to MPEG-G” for further clarification.

Reply 2.3 — Thank you for your suggestion regarding the presentation of the MPEG-G standard in our paper. The MPEG-G standard is indeed comprehensive, comprising several parts that address different facets of genomic data representation, storage, and processing. These are:

1. Part 1: Transport and Storage of Genomic Information
2. Part 2: Coding of Genomic Information
3. Part 3: Metadata and Application Programming Interfaces
4. Part 4: Reference Software (decoder)
5. Part 5: Conformance Testing
6. Part 6: Coding of Genomic Annotations

In our paper, we primarily focus on aspects defined in Part 2, which are directly relevant to the processes implemented in Genie. To improve clarity, we have added an explanatory section in the introductory part of the paper. This section outlines the relationship between Genie and MPEG-G, particularly highlighting the relevance of Part 2.

The standard currently comprises six parts (1. Transport and Storage of Genomic Information, 2. Coding of Genomic Information, 3. Metadata and Application Programming Interfaces, 4. Reference Software, 5. Conformance Testing, 6. Coding of Genomic Annotations). The processes implemented in the Genie encoder are mostly based on the second part of the standard. For more information about the standard and its features beyond the scope of this paper, we refer to the MPEG-G introductory paper (20).

While we initially considered the idea of incorporating a diagram of the MPEG-G standard, we concluded that it might not significantly enhance the reader’s comprehension beyond what is already depicted in Figure 1 of our paper. The encoding process demonstrated in Genie essentially mirrors the decoding processes outlined in MPEG-G Part 2.

At your suggestion, we have referenced the paper “An Introduction to MPEG-G,” which provides comprehensive insights into the standard, including aspects that extend beyond the scope of Genie.

Comment 2.4 — The comparison of genie with FASTA/Q compressors should include FaStore, SCALCE, and Leon.

Reply 2.4 — Thank you for your suggestion regarding the inclusion of FaStore, SCALCE, and Leon in our comparison of FASTQ compressors. We appreciate the importance of a comprehensive and current benchmark in evaluating Genie’s performance. However, based on an in-depth analysis of published results, we came to the conclusion that these tools may no longer represent the state of the art:

1. Published results for the evaluation of FaStore (4) show that FaStore delivers superior average compression ratios compared to Leon and SCALCE (around 4 percentage points w.r.t. both tools) as well as higher decompression speeds (around twice the speed w.r.t. to Leon and a 5-10 percent increase w.r.t. SCALCE).
2. Published results for SPRING show superior compression ratios (1.1x to 1.6x depending on the dataset) compared to FaStore while using at least “comparable computational resources” (9).

Therefore, we decided to include other, more recent compressors that improve upon Spring compression ratio instead: PgRC (1) and Mstcom (2). We also included Genozip (3) which has a mode for the compression of FASTQ data comparable to the low-latency compression mode of Genie.

Comment 2.5 — It is unclear how low latency, local assembly, and reference-based encoding is implemented in the paper. It is dependent or independent of the parameters in the genie architecture described in figure 1a?

Reply 2.5 — Thank you for your comment. Genie’s data compression approach, aligned with the MPEG-G standard, primarily involves comparing genomic records to a reference sequence, thus storing only the differences rather than the entire sequence. The encoding modes — low latency, local assembly, global assembly, and reference-based — determine how this reference sequence is obtained:

1. Reference-based Encoding: Utilizes an **external reference** sequence from a FASTA file for comparing aligned genomic records. This mode is efficient for compressing aligned records in a SAM file, leveraging the pre-existing alignment to the reference for compression.
2. Local and Global Assembly: Both modes generate a **computed reference** sequence from the genomic records themselves, without external data.
 - (a) Local Assembly: Constructs a reference by performing a majority vote on mapped bases at each genomic locus in aligned records.
 - (b) Global Assembly: Creates a pseudo-reference sequence from unaligned records by identifying similar and overlapping nucleotide sequences. This sequence has no biological meaning but serves as a basis for compression.
3. Low Latency Encoding: For unaligned records, this mode encodes the nucleotide sequence directly, with **no reference** comparison. It prioritizes speed over compression efficiency.

The choice of encoding mode depends on the alignment status (aligned/unaligned) of the genomic records and the availability of a reference sequence. For aligned records, reference-based encoding is used if an external reference is available; otherwise, local assembly is the default. For unaligned records, the user can choose low latency encoding; if not selected, global assembly is applied by default.

After determining the differences to the reference sequence, the subsequent processing and entropy encoding are generally independent of the chosen reference strategy. However, the encoding mode may

influence the statistical properties of the data streams. The optimized parameters discussed later in the paper pertain to this subsequent processing phase.

To clarify these concepts, we have added an additional explanation to the manuscript, delineating the distinction between the reference mode and other parameters in Genie's architecture:

The Genie encoding process involves parameters that fall into two distinct categories. As described above, the first category encompasses parameters for the different encoding strategies responsible for generating descriptor sequences. The selection of these parameters is inherently determined by the characteristics of the data, such as whether the records are aligned or unaligned and the lengths of the nucleotide sequences within those data sets. Since these characteristics are directly data-dependent, optimization in this context is not necessary. The second category pertains to parameters associated with subsequent (optional) transformations of previously generated descriptor sequences and the final entropy encoding stage. Unlike the first category, these parameters are not rigidly dictated by the simple properties of the records. Instead, they provide flexibility and room for optimization to adapt to the statistical properties of the descriptor streams. Therefore, *optimized parameters* in the following context refers to the choice of transformations (sequence transformation, subsymbol transformation) and parameters of the entropy coding (binarization, context size, etc.) after descriptor subsequences have been created (see the "Descriptor Subsequences" arrow in figure 1a).

Comment 2.6 — In the first paragraph on page 4 where the mention of "an optimized set of parameters" appears, it should be mentioned what these parameters are referring to.

Reply 2.6 — Thank you for your comment. As discussed in reply 2.5, these optimized parameters correspond to the parameters of the transformations and entropy coding processes after descriptor subsequence generation. This involves activation/deactivation of transformations as well as parameters for the CABAC encoding such as the context size. We think that the additions we inserted into the paper for comment 2.5 address this comment as well.

Comment 2.7 — Why was Genie only tested on 8 datasets from the MPEG-G genomic information database and not all of them. There should be some justification for this mentioned in the paper because it might give the reader the wrong impression with respect to the experimental portion of the paper.

Reply 2.7 — Thank you for your comment. We agree to these concerns and have therefore significantly extended our experiments. Instead of using just 8 datasets, we now tested all compression tools with all items in the MPEG-G genomic database. We refer to reply 1.2 where we explained the new experimental setup and datasets in detail.

Comment 2.8 — Again on page 5, the last paragraph, it should be noted what the parameter set is referring to.

Reply 2.8 — Thank you for your feedback regarding the last paragraph on page 5. To clarify, we have added an additional sentence specifying the nature of the 'parameter set', similar to a summary of the explanation we added in response to comment 2.6.

Note that on datasets that have not been used previously for the optimization of transformation and entropy encoding parameters as described above, Genie still achieves acceptable

compression ratios if the underlying statistical properties and sequencing technology are not completely different.

Comment 2.9 — On page 8, in filtering out of data, why are optional tags removed? Does that leave out important information? Also it is unclear whether other approaches remove these items.

Reply 2.9 — Thank you for your question about the removal of optional tags in SAM files during our data filtering process. Optional tags in SAM files provide additional annotations for each record, such as details from base calling, alignment processes, or information about mate records in paired sequencing. The relevance of these tags varies:

1. Some tags contain information that can be recalculated, such as “Edit distance to the reference.”
2. Certain tags are not necessary in the MPEG-G format. For example, SAM format stores records, even those forming a pair, as independent lines in the SAM file. The SAM file is often sorted by mapping position, which means that a record's mate does not have to directly follow the record. SAM files may use optional tags for details about mate records to avoid searching them in the file for retrieval of certain information. MPEG-G, however, stores mate records together in a joint data structure, eliminating the need for such searches.
3. There are tags with unrecoverable information, like “Base modifications.”

The necessity of these tags largely depends on the specific downstream tasks and their requirements. The other methods include SAM tags in their bitstreams. Optional tags are not a core feature in MPEG-G, but are regarded as part of metadata (reflected by the fact that the encoding of tags is specified in Part 3 “Metadata and Application Programming Interfaces” of the standard and not in Part 2 “Coding of Genomic Information”). Therefore, the implementation of optional tags presents a significant investment of effort in comparison to the utility they offer. Given this disproportionate effort-to-benefit ratio, we decided to allocate our resources towards the implementation of other features instead that promise greater impact on the usefulness of Genie.

To ensure a fair comparison in our experiments with other approaches, which do not remove these tags, we filtered them out from all datasets before conducting the experiments. This step was taken to maintain consistency across all tested methods and provide an equitable basis for comparison. To clarify the role of optional tags, we added this explanation to the manuscript:

Optional tags in SAM files provide additional annotations on a per-record level, such as details from base calling, alignment processes, or information about mate records in paired sequencing. The usefulness of these tags largely depends on the specific downstream tasks and their requirements.

Comment 2.10 — On page 11, why is $i < 10$? In other words, how come there are 9 possible access units in file f ? 6 records classes, 2 possibilities for number of nucleotide sequences, and two possibilities for presence or absence of a reference sequence. A supplementary note on what the 9 possible access units are would be nice as the reader might get confused about this number.

Reply 2.10 — Thank you for your comment. The choice of the parameter i does not represent the total number of possible access units in a file, but it is instead a pragmatic decision for sampling size.

In our methodology, we aim to optimize parameters based on an initial subset of access units in a file, working under the assumption that these units are representative of the statistical properties of the entire file. Our analysis indicated that the first 10 access units provide a balanced approach, offering a manageable computational complexity while also encompassing most types of access units that significantly appear in the file.

To clarify this in the paper, we have included this brief explanation of our approach to selecting this sample size.

Ten access units were chosen as a sample for the full file, since we found that this sample size provides a balanced approach, offering a manageable computational complexity while also already encompassing most types of access units that appear in the file in significant amounts.

Additionally, to address any potential confusion, we have added a paragraph detailing the various types of possible access units in the section “Regrouping of Records”.

With 5 classes of records (P, N, M, I, HM), 2 pairing configurations (paired / unpaired) and n reference sequences, there are at most $5 \cdot 2 \cdot n$ buffers for aligned records. Additionally, with one class (U), 2 pairing configurations (paired / unpaired) and no references, there are at most 2 buffers for unaligned sequences. However, in most datasets, not all of these record types actually occur, thus some of these buffers typically remain unused.

This distinction between the sample size i and the actual number of potential access units will help readers better understand the rationale behind our experimental setup.

Comment 2.11 — Should there be a disclosure statement similar to that in doi/10.1146/annurev-biodatasci-072018-021229 stating Mikel Hernaez and Jan Voges have contributed to the development of MPEG-G standard?

Reply 2.11 — Thank you for raising your concerns. As you correctly noted, all authors of this paper have indeed contributed to various parts of the MPEG-G standard. In line with best practices and to ensure complete transparency, we have decided to include a disclosure statement in our paper. The disclosure will be placed towards the end of the paper, clearly informing readers of our roles in the development of the MPEG-G standard:

Declarations

All authors have contributed to the development of the MPEG-G standard ISO/IEC 23092.

Reviewer 3

Comment 3.1 — It could be interesting to also report the single-threaded performance in addition to the multi-threaded performance.

Reply 3.1 — Thank you for suggesting the inclusion of single-threaded performance data. We acknowledge that evaluating single-threaded performance can offer valuable insights into the fundamental efficiency of the compression methods.

To address this, we have included single-threaded compression and decompression times for two selected files (one aligned and one unaligned). It's important to note that single-threaded runtimes are considerably longer for most methods compared to multithreaded execution, and we only have limited computational resources available. For this reason, we have limited the single-threaded analysis to only a single dataset per method, rather than the full range used in other experiments. We believe that this is already sufficient for a qualitative comparison on how the different approaches benefit from multiple CPU cores. Additionally, as the compression ratio is not affected by threading, we only report runtimes for single-threaded operations, with the understanding that the compressed sizes remain consistent with those observed in the multithreaded scenarios. To refer to these results, we added the following to the main text of the manuscript:

We also recorded the single-threaded performance of all tools for two selected data items to provide a qualitative comparison of the impact of multithreading. These results are reported in the extended data figure 4a. The benefit of multiple threads differs notably between tools. A plausible reason could be a difference in how coding speed is limited by CPU resources in comparison to I/O bandwidth.

Figure 4: **Extended experimental data.** a, Ratio between singlethreaded and multithreaded (8 threads) encoding/decoding times for item 01-1 (unaligned data) and item 9-2 (aligned data), respectively.

Comment 3.2 — Another interesting discussion would be to better elaborate on the overall loss or gain in coding efficiency for using an optimal parameter set for each access unit versus using one optimized parameter set for all access units.

Reply 3.2 — Thank you for your comment. To add some more analysis regarding this topic, we reevaluated the optimization of our parameter sets. We compared the case of global optimization of the sequence transformation with one global parameter set to the local optimization with a separate parameter set per access unit. Our results indicate that the loss with this simplification results in a minimal loss far below 1 % for access units inside the sample data used for optimization. We added the following text in the methods section to reference our table of results:

As an additional experiment, we also compared this approach of using only one, globally optimized parameter set to using an individually optimized parameter set per access unit under the aspect of the sequence transformation. Our results indicate that the loss in compression ratio is insignificant ($< 1\%$) when choosing the simpler, global approach. The numbers for this experiment are presented in table 4 in the appendix F.

Encoding	Item	Compressed Size (Bytes)		Size Ratio
		Dynamic Conf.	Static Conf.	
Low Latency	01-1	70092398	70160007	1,001
Global Assembly	01-1	157331725	157552232	1,001
Low Latency	32	55909170	55927424	1,000
Global Assembly	32	50476217	50595832	1,002
Local Assembly	02-1	109425305	109557150	1,001
Reference-Based	02-1	101174649	101275082	1,001
Local Assembly	37	44190804	44338928	1,003
Reference-Based	37	37329463	37440621	1,003

Table 4: The compressed size of the first ten access units of selected items when compressed using a constant. Compared are the sizes for a globally optimized, constant parameter set versus dynamic parameter set that is optimized individually for each access unit. The last column is the ratio between both sizes.

Comment 3.3 — It could be also interesting to add results on the performance of long-read (nanopore) data

Reply 3.3 — Thank you for your suggestion to include performance results on nanopore data. In response, we have significantly expanded our dataset by incorporating the full MPEG-G genomic information database (GIDB), which includes a diverse array of datasets from various sequencing technologies. In particular, we have added both an unaligned and aligned human dataset from ONT PromethION, as well as *E. coli* datasets from ONT MinION. Additionally, the extended experiments contain a dataset generated using Pacific Biosciences (PacBio) technology, another source of long-read data.

One notable observation from our expanded analysis is that most sorting / assembly-based compression methods struggled with unaligned PromethION data, with the notable exceptions of Genie in its low-latency mode and Genozip. This finding underscores the unique challenges and considerations when compressing long-read sequencing data.

We have incorporated these results into the updated box plots in Figure 2 and in Figure 3. We refer to reply 1.2 for more details about the extended experiments and the updated figures.

Comment 3.4 — Another interesting result would be to evaluate the trade-off between access unit size and compression ratio.

Reply 3.4 —Thank you for your suggestion to evaluate the trade-off between access unit size and compression ratio.

To address this, we conducted a focused analysis using a subset of records from a dataset within the MPEG-G genomic information database (GIDB). Our investigation specifically looked at how varying the size of access units impacts the compression ratio.

As anticipated, we observed an improving compression ratio with a saturation effect as access unit size increased. This trend was particularly pronounced in the reference-based encoder, which showed a more significant benefit from larger access units compared to other encoding strategies. This is likely due to the reference-based encoder's ability to produce smaller descriptor sequences before entropy encoding. It leverages an external reference, reducing the need to store either a computed reference or the raw bases. Consequently, the proportion of overhead relative to the total compressed data becomes more significant for the same access unit size, when compared to the other encoding methods that do not use an external reference.

We have included a new figure 3a in the manuscript to illustrate these results, providing a visual representation of how the compression ratio evolves with changes in access unit size across different encoding strategies. Additionally, we reference those results in the main text:

In Genie, the size of an access unit is a trade-off between granular access capabilities and compression ratio, as compression overhead as well as access granularity increase with smaller access units. Figure 3a shows the impact of access unit size on compression ratio in Genie. The compression ratio saturates as block sizes increase. The reference-based encoding mode (Genie-Ref) benefits most from larger block sizes, with a 32% decrease in compressed size between 2^9 and 2^{18} records per access unit. The reason is likely that most of the record information is already located in the external reference, leading to smaller descriptor streams, thus more overhead for the same number of records compared to other encoding modes. For all other encoding modes, the impact of larger access units is similarly small, at approximately 23%.

Figure 3: **Additional experimental data.** a, Compressed size of the first 2^{18} records in item 38 (Ref, LA) or items 49 + 50 (GA, LL), respectively, depending on the encoding block size (smallest block size: 2^9 records, largest block size 2^{18} records).

Comment 3.5 — It would also be worth mentioning in the text that the more recent edition of the standard also include a set of low complexity entropy coders that so far are not supported by the Genie implementation, but that could be used in future to optimize speed performance for some type of data such as quality values for instance.

Reply 3.5 — Thank you for your suggestion. We have now incorporated a specific mention in our paper, particularly in the sections discussing compression and decompression speeds. Some random samples that we have benchmarked have shown that a significant portion of Genie’s runtime is used for the CABAC entropy encoding process (up to 85 percent, according to a quick, non-systematic callgrind (17) analysis that we conducted). Given this observation, we see the potential for significant speed improvements in future versions of Genie by adopting these low-complexity codecs. We believe this addition to the paper provides readers with a possible pathway for the future development of Genie.

For aligned records, BAM, CRAM, and Genozip show comparable encoding speeds, while DeeZ encoding takes around 5 times as long. Genie-Ref and Genie-LA encoding runtimes are similar to each other, but approximately 3 to 4 times longer than DeeZ. Further analysis shows that over 80% of the Genie runtime for aligned data is spent for the entropy encoding using CABAC. A plausible explanation for the high computational complexity of CABAC is the bit-based compression approach it employs, while most other entropy codecs process larger symbols. However, this issue could be resolved in future versions of Genie, as upcoming

editions of the MPEG-G standard promise faster entropy encoding by including a number of low-complexity entropy codecs, such as BSC (13), LZMA (14), or Zstandard (15).

References

- [1] T. M. Kowalski and S. Grabowski, "PgRC: pseudogenome-based read compressor," *Bioinformatics*, vol. 36, no. 7, pp. 2082–2089, 12 2019. [Online]. Available: <https://doi.org/10.1093/bioinformatics/btz919>
- [2] Y. Liu and J. Li, "Hamming-shifting graph of genomic short reads: Efficient construction and its application for compression," *PLOS Computational Biology*, vol. 17, no. 7, pp. 1–16, 07 2021. [Online]. Available: <https://doi.org/10.1371/journal.pcbi.1009229>
- [3] D. Lan, R. Tobler, Y. Souilmi, and B. Llamas, "Genozip: a universal extensible genomic data compressor," *Bioinformatics*, vol. 37, no. 16, pp. 2225–2230, 02 2021. [Online]. Available: <https://doi.org/10.1093/bioinformatics/btab102>
- [4] L. Roguski, I. Ochoa, M. Hernaez, and S. Deorowicz, "FaStore: a space-saving solution for raw sequencing data," *Bioinformatics*, vol. 34, no. 16, pp. 2748–2756, 03 2018. [Online]. Available: <https://doi.org/10.1093/bioinformatics/bty205>
- [5] F. Hach, I. Numanagić, C. Alkan, and S. C. Sahinalp, "SCALCE: boosting sequence compression algorithms using locally consistent encoding," *Bioinformatics*, vol. 28, no. 23, pp. 3051–3057, 10 2012. [Online]. Available: <https://doi.org/10.1093/bioinformatics/bts593>
- [6] G. Benoit, C. Lemaitre, D. Lavenier, E. Drezen, T. Dayris, R. Uricaru, and G. Rizk, "Reference-free compression of high throughput sequencing data with a probabilistic de bruijn graph," *BMC Bioinformatics*, vol. 16, p. 288, Sep. 2015.
- [7] Ł. Roguski and S. Deorowicz, "DSRC 2 — industry-oriented compression of FASTQ files," *Bioinformatics*, vol. 30, pp. 2213–2215, 2014.
- [8] D. Jones, W. Ruzzo, X. Peng, and M. Katze, "Compression of next-generation sequencing reads aided by highly efficient de novo assembly," *Nucleic acids research*, vol. 40, p. e171, 2012.
- [9] S. Chandak, K. Tatwawadi, I. Ochoa, M. Hernaez, and T. Weissman, "SPRING: a next-generation compressor for FASTQ data," *Bioinformatics*, vol. 35, pp. 2674–2676, 2019.
- [10] F. Hach, I. Numanagic, and S. Sahinalp, "DeeZ: reference-based compression by local assembly," *Nature Methods*, vol. 11, pp. 1082–1084, 2014.
- [11] J. Bonfield, "CRAM 3.1: advances in the CRAM file format," *Bioinformatics*, vol. 38, pp. 1497–1503, 2022.
- [12] MPEG. (2018) Mpeg-g genomic information database. [Online]. Available: <https://mpeg.chiariglione.org/standards/MPEG-G/genomic-information-representation/MPEG-G-genomic-information-database>
- [13] I. Grebnov. (2009) libbsc. [Online]. Available: <http://libbsc.com/>
- [14] I. Pavlov. (1998) Lzma. [Online]. Available: <https://7-zip.org/sdk.html>
- [15] Y. Collet and M. Kucherawy, "Zstandard Compression and the application/zstd Media Type," RFC 8478, Oct. 2018. [Online]. Available: <https://www.rfc-editor.org/info/rfc8478>
- [16] F. Pedregosa and P. Gervais. (2011) Memory profiler. [Online]. Available: <https://pypi.org/project/memory-profiler/#history>
- [17] J. Seward. (2000) Valgrind. [Online]. Available: <https://valgrind.org/>
- [18] GA4GH. (2023) Cram format specification (version 3.1). [Online]. Available: <https://samtools.github.io/hts-specs/CRAMv3.pdf>
- [19] I. Numanagić, J. K. Bonfield, F. Hach, J. Voges, J. Ostermann, C. Alberti, M. Mattavelli, and S. C. Sahinalp, "Comparison of high-throughput sequencing data compression tools," *Nature Methods*, vol. 13, no. 12, pp. 1005–1008, Dec 2016. [Online]. Available: <https://doi.org/10.1038/nmeth.4037>
- [20] J. Voges, M. Hernaez, M. Mattavelli, and J. Ostermann, "An introduction to MPEG-G: The first open ISO/IEC standard for the compression and exchange of genomic sequencing data," *Proceedings of the IEEE*, vol. 109, pp. 1607–1622, 2021.

REVIEWERS' COMMENTS:

Reviewer #1 (Remarks to the Author):

My previous concerns have been addressed.

Reviewer #3 (Remarks to the Author):

The authors have satisfactorily addressed all my previous requests and have completed the manuscript with additional experimental results. I have no other requests and I recommend the publication of the paper in its current form.